# Alpha-1 Acid Glycoprotein Reduction Differentiated Recovery from Remission in a Small Cohort of Cats Treated for Feline Infectious Peritonitis

**DOI:** 10.3390/v14040744

**Published:** 2022-04-01

**Authors:** Diane D. Addie, Carla Silveira, Charlotte Aston, Pauline Brauckmann, Johanna Covell-Ritchie, Chris Felstead, Mark Fosbery, Caryn Gibbins, Kristina Macaulay, James McMurrough, Ed Pattison, Elise Robertson

**Affiliations:** 1Independent Researcher, 64470 Etchebar, France; 2Independent Researcher, London, UK; carla.lambisgoia@gmail.com; 3Independent Researcher, Melton Mowbray, UK; meltonvets@gmail.com; 4Independent Researcher, Östersund, Sweden; pauline.brauckmann1@gmail.com; 5Independent Researcher, Maidstone, Kent, UK; johannacovell@hotmail.com; 6Independent Researcher, Bracknell, UK; chris.felstead@medivet.co.uk; 7Independent Researcher, Maidstone, UK; mark.fosbery@newnhamvets.com; 8Independent Researcher, Hampstead, UK; hampstead@zasmanvet.co.uk; 9Independent Researcher, Ayr, UK; kristinamac@mac.com; 10Independent Researcher, Manchester, UK; james.mcmurrough@vets-now.com; 11Independent Researcher, Exeter, UK; ed@cityvets.co.uk; 12Independent Researcher, Brighton, UK; felinevet369@gmail.com

**Keywords:** feline coronavirus, feline infectious peritonitis, FIP, treatment, alpha-1 acid glycoprotein, AGP, acute-phase protein, interferon omega, antiviral, meloxicam

## Abstract

Feline infectious peritonitis (FIP) is a systemic immune-mediated inflammatory perivasculitis that occurs in a minority of cats infected with feline coronavirus (FCoV). Various therapies have been employed to treat this condition, which was previously usually fatal, though no parameters for differentiating FIP recovery from remission have been defined to enable clinicians to decide when it is safe to discontinue treatment. This retrospective observational study shows that a consistent reduction of the acute phase protein alpha-1 acid glycoprotein (AGP) to within normal limits (WNL, i.e., 500 μg/mL or below), as opposed to duration of survival, distinguishes recovery from remission. Forty-two cats were diagnosed with FIP: 75% (12/16) of effusive and 54% (14/26) of non-effusive FIP cases recovered. Presenting with the effusive or non-effusive form did not affect whether or not a cat fully recovered (*p* = 0.2). AGP consistently reduced to WNL in 26 recovered cats but remained elevated in 16 cats in remission, dipping to normal once in two of the latter. Anaemia was present in 77% (23/30) of the cats and resolved more quickly than AGP in six recovered cats. The presence of anaemia did not affect the cat’s chances of recovery (*p* = 0.1). Lymphopenia was observed in 43% (16/37) of the cats and reversed in nine recovered cats but did not reverse in seven lymphopenic cats in the remission group. Fewer recovered cats (9/24: 37%) than remission cats (7/13: 54%) were lymphopenic, but the difference was not statistically different (*p* = 0.5). Hyperglobulinaemia was slower than AGP to return to WNL in the recovered cats. FCoV antibody titre was high in all 42 cats at the outset. It decreased significantly in 7 recovered cats but too slowly to be a useful parameter to determine discontinuation of antiviral treatments. Conclusion: a sustained return to normal levels of AGP was the most rapid and consistent indicator for differentiating recovery from remission following treatment for FIP. This study provides a useful model for differentiating recovery from chronic coronavirus disease using acute phase protein monitoring.

## 1. Introduction

Feline coronavirus (FCoV) is a positive strand RNA virus belonging to the order Nidovirales, genus Coronavirus, and family *Coronaviridae*, subfamily alphacoronavirus. The FCoV species is further divided into two types: I and II, the first type being wholly feline, and type II FCoV arising from recombination events with canine coronavirus [1,2].

Feline infectious peritonitis (FIP) is an immune-mediated perivascular pyogranulomatous [3] disease which affects 5–10% of cats infected with FCoV [4]. FIP can present acutely, with effusions in one or more body cavity, or as a disease of chronic inflammation, with cachexia and variable organ damage depending on the sites of pyogranuloma formation.

The initial target for FCoV is the epithelial cells of the small intestine [5]; from there, macrophages engulf the virus and transport it to the mesenteric lymph nodes (MLN), and a brief systemic phase of infection follows when the monocyte is the target cell.

FCoV is highly prevalent in the cat population, but FIP is relatively rare. There are two theories for why FIP occurs; the first is the internal mutation theory, postulating that some mutation is required in an otherwise benign virus to permit replication in monocytes [6]. Spike gene mutations known as M1058L or S1060A were considered to be virulence markers and evidence for FIP diagnosis [7,8], though it has been argued that these mutations are only indicative of the systemic spread of the virus [9], and current expert opinion is that they are not useful diagnostic tests [10,11,12,13]. The second theory is the circulating virulent–avirulent theory which states that distinct benign and virulent, FIP-causing strains circulate in the feline population [14,15]. Whichever theory is true, the rôle of virus load is critical but often overlooked. Pedersen & Black found that kittens infected with a small amount of virulent laboratory strains of FCoV did not develop FIP. At medium doses, FIP was sporadic, but higher doses were almost always lethal [16]. This pattern of sporadic deaths and occasional FIP outbreaks reflects what is seen in real life, where outbreaks of FIP occur in environments where a high environmental virus burden would be expected, such as in rescue or breeding catteries [2,15,17,18,19]. The stress of entering a rescue cattery increases virus load in some individual cats [20], and stress is reported to precede the onset of FIP [21,22].

Progression to FIP is primarily driven by viral replication [23]. The ability of the virus to replicate to high titres in monocytes depends upon the host rather than the virus strain. Some host monocytes do not permit viral replication regardless of how virulent is the strain of FCoV [24].

In FIP, coronavirus-infected monocytes/macrophages release matrix metalloproteinase-9 (MMP-9) [3]. MMP-9 attacks the collagen IV in the vascular basal lamina allowing extravasation of FCoV-infected monocytes (which differentiate into macrophages) [3] and leakage of protein-rich plasma (i.e., a modified transudate) into body cavities such as the peritoneal or pleural cavities, pericardial sac [13,15,25], or scrotal sac [26] in effusive (or wet) FIP. Plasma leakage into the brain’s ventricles leads to hydrocephalus [27,28]. B cells gradually replace macrophages in more chronic perivascular pyogranulomata [29] typical of non-effusive FIP. FCoV-infected macrophages release a storm of cytokines and chemokines, including tumor necrosis factor-alpha (TNF-α) [3,30,31]^.^ and interleukin-6 (IL-6) [32].

TNF-α is a major contributor to the inflammatory response and pathogenesis of FIP. FCoV-infected cells release a substance that causes apoptosis in nearby lymphocytes [33]: the mystery substance is probably TNF-α [30]: around 50% of cats with FIP are lymphopenic [22,34,35]. In an experimental infection of specific pathogen-free cats, lymphopenia began around 2–4 weeks post-infection and correlated with the disease course: the earlier post-infection that lymphopenia occurred, the more rapid the progression of the disease [36].

Cats with FIP typically have the non-regenerative normocytic normochromic anaemia of chronic disease [13]. A study of 51 cats with FIP found that a decreasing red blood cell count was a significant prognostic indicator: a packed cell volume of under 20% (along with other prognostic indicators such as elevated bilirubin and aspartate aminotransferase, low potassium, and sodium) heralded a survival of less than three days [34].

IL-6 stimulates hepatocytes to release acute phase proteins [32], such as alpha-1 acid glycoprotein (AGP) and serum amyloid A (SAA). AGP levels rise in all acute viral and bacterial infections [37,38], *Mycoplasma hemofelis* infection [39], and after trauma. AGP was first reported to be elevated in FIP cases in 1997 [40], but it also rises transiently post FCoV-infection even in cats who do not develop FIP [41,42]. Raised AGP is superior to biopsy histopathology [43], SAA, and haptoglobin [44] in differentiating FIP from similarly-presenting cases. In a cat with an effusion, AGP levels above 1550 μg/mL in cats were 93% specific for FIP [44]. In non-effusive FIP, AGP levels tend to be above normal, but often not markedly so [Addie, personal observation], presumably because non-effusive FIP is a more chronic presentation. A normal AGP level usually rules out FIP [11]. Monitoring AGP is an accurate predictor of survival in humans with sepsis [45].

Many treatments, including prednisolone, feline interferon omega (rFeIFN omega) [46], polyprenyl immunostimulant [47], meloxicam [48], and most recently specific antivirals [49,50,51,52,53] have been used. Those treatments involved in the present study are detailed in Table 1 and Table 2.

There are three possible outcomes following the diagnosis of FIP and treatment: death due to FIP-related causes; total recovery; or remission, defined as an intermediate stage between cure and death and carrying the specter of relapse. This latter state is a source of considerable stress for cat guardians, so it would bring great reassurance to people to know that their cats are cured of FIP rather than being in remission.

Veterinary surgeons seek a criterion to be assured that the cat has recovered to know when to cease treatment. Markers for poor prognosis in FIP have been well defined [34], yet no guidelines have been established to define recovery from FIP, probably because, until recently, it was widely assumed that recovery from FIP could not occur.

Prior to the advent of antivirals specific to coronavirus, the average survival time of cats with effusive FIP was 21 days [34], while cats with non-effusive FIP tended to survive longer, on average 38 days [34]. No survival time point after which a cat can be said to have recovered from FIP has been established. In their report of their seminal study of successful treatment of FIP with an antiviral drug, Pedersen et al. wrote: “It raises the question of how long remission must be sustained to declare the disease cured, rather than in a sustained remission.” [49] The purpose of this paper was to attempt to answer that question, but as opposed to duration of survival, we found that a return of elevated AGP to normal levels is a rapid and definitive marker for recovery from FIP.

## 2. Materials and Methods

### 2.1. Selection of Cases

This study retrospectively evaluated the medical records of 42 cats diagnosed with FIP between January 2004 and March 2021. Cats were selected for this study if sequential laboratory test results were available, including AGP measurements, a reasonable certainty that the FIP diagnosis was correct, and whose survival outcomes were known.

### 2.2. FIP Diagnosis

Diagnosis of FIP was by histopathology, immunohistochemistry (IHC); detection of FCoV RNA as demonstrated by RT-PCR in effusions, [54,55,56] or mesenteric lymph node (MLN) FNA [57]; messenger RNA (mRNA) [58,59] or three prime–untranslated region (3′-UTR) RNA [60,61] in peripheral blood mononuclear cells (PBMC); FCoV spike gene mutations M1058L, S1060A [7,8]; and FIP profile based on an algorithm consisting of history, clinical signs and clinicopathological abnormalities considered consistent with FIP, including high FCoV antibody titre, raised AGP, hyperglobulinaemia, anaemia, and lymphopenia [62,63].

### 2.3. Treatment

Treatment choice was at the discretion of the cat’s attending veterinary surgeon or, more recently, the cat’s guardian, with two exceptions. Previously, prednisolone was considered an integral part of FIP treatment [64], but systemic corticosteroids were subsequently found to decrease survival time in cats that were being treated concurrently with polyprenyl immunostimulant (PI, VetImmune, Sass & Sass, Inc., Oak Ridge, TN, USA) [47]. Consequently, in 2017 one author (DDA) recommended stopping (or not commencing) the use of corticosteroids and using meloxicam instead (after a suitable wash out period and provided blood pressure and kidney function were normal). The second exception was that clients whose cats were treated with adenosine nucleoside analogue drugs were recommended to supplement with S-adenosyl-L-methionine (SAMe) and silybin phosphatidycholine complex, (Denamarin, Nutramax Laboratories, Inc., Lancaster, SC, USA) for liver support. 

### 2.4. Outcomes: Death, Recovery, or Remission

Recovery was defined as the resolution of all the clinical signs attributable to FIP and reversal of anaemia, lymphopenia, and hyperglobulinaemia, if present. AGP was not included in the decision to classify a cat as recovered or only in remission because the purpose of this study was to assess whether or not AGP reduction was an indicator of clinical recovery. Clinical scores are shown in Table 3 and Table 4.

### 2.5. AGP

AGP was measured by enzyme-linked immunosorbent assay (ELISA) (Avacta Animal Health, Wetherby, Yorkshire, UK) at the University of Glasgow Veterinary Diagnostic Services (VDS) in all cases except seven: the samples of Basil 1; Amy; Brook; Bugsy; Daisy; Roxanne and Pip were measured by radial immunodiffusion as previously described [40].

The maximum value given on the *y* axis of Figure 1 and Figure 2 was set at 5000 μg/mL to maintain the legibility of the graphs. The minimum ELISA cut-off was 300 μg/mL, and ELISA results reported as “<300” were plotted as 300.

### 2.6. Statistics

Due to the small sample size, Fisher’s exact test was used with significance set at <0.05.

## 3. Results

### 3.1. FIP Diagnosis

The laboratory records of effusive (*n* = 16) and non-effusive (*n* = 26) FIP cases were reviewed. As shown in Table 1 and Table 2, the diagnosis of FIP was by histopathology (*n* = 13), of which two cases had positive immunohistochemistry (IHC). However, the histopathology of biopsies was inconclusive in four cases (Elmo, Harry, Pharaoh, Ragamuffin) and was reported non-specifically as “pyogranulomatous inflammation”.

FIP diagnosis was by detection of FCoV RNA as demonstrated by RT-PCR in effusions [54,55,56,60] (*n* = 10), or mesenteric lymph node (MLN) FNA [57] (*n* = 10); messenger RNA (mRNA) [58,59] or 3′-UTR [60,61] in PBMC (*n* = 4). One of two cats in which the mutation tests [7,8] were performed was positive for the M1058L mutation, and neither cat was positive for the S1060A mutation. Some cats fulfilled multiple diagnostic criteria for FIP: seven cats had both positive FCoV RT-PCR tests and histopathology. All 42 cats were positive for at least some components of an FIP profile consisting of typical clinical signs, FCoV antibody titre, raised AGP, hyperglobulinaemia, anaemia, and lymphopenia (Table 3 and Table 4). In 11 cats, the FIP profile was the sole evidence for its diagnosis, although a positive response to an anti-coronavirus drug was thought to corroborate the FIP diagnosis. AGP was raised in all cats except two, where a pre-treatment test was not done.

**Table 1 viruses-14-00744-t001:** How FIP was diagnosed, survival time, and treatment details of 26 cats who recovered.

	Cat	FIP Presentation	How FIP Diagnosed	Survival in Years or Months	Time to Normal AGP	Treatments	Prednisolone/Corticosteroids
1	Basil 1	Non-effusive (icterus)	mRNA RT-PCR on PBMC positive thrice.	11y † (Died of chronic kidney disease aged 15y.)	<22d	5 × 10^4^ units of rFeIFN-ω (Virbagen Omega, Virbac, Nice, France) *per os* q24h for 13m until FCoV antibody titre reduced from >1280 to <1:10.	2 mg/kg q24h for 7d then 1 mg/kg for 7d
2	Boris	Non-effusive but initially effusive FIP suspected	RT-qPCR on MLN FNA C*_T_* 33.7. RT-qPCR on ascites negative.	>2.2y	<3m	1 MU units rFeIFN-ω *per os* q24h. PI 3 mg/kg *per os* q72h. Weekly cobalamin injections. Effusion was negative on RT-PCR and was found to be due to cardiomyopathy.	No
3	Mars	Non-effusive	RT-qPCR on MLN FNA C*_T_* 30.	>5.5y	<6m	PI 3 mg/kg twice per week.	No
4	Chester	Effusive (pleural effusion)	RT-qPCR on pleural effusion C*_T_* 34.	>3.1y	<8m	1 MU/kg rFeIFN-ω s/c q48h reduced to q4d then 1 × 10^5^ units *per os* q24h for 28m. Meloxicam. SAMe (Denamarin^®^, Nutramax Laboratories, Inc., Lancaster, SC, USA) Cobalamin (Cobalaplex, Protexin^®^ Veterinary, Somerset, UK). Tramadol hydrochloride (Bova Specials, London, UK) 10 mg/cat q24h *per os.* [48] Liquorice tea was attempted but cat did not like it.	1 mg/kg q12h for 2w, weaned off for another 3w and replaced by meloxicam
5	Amy	Non-effusive	FIP profile.	>1.3y	<4m	1 MU units rFeIFN-ω s/c q48h.	Sliding doses
6	Brook	Non-effusive	FIP profile.	>1.1y	<41d	Unknown.	Unknown
7	Basil 2	Effusive (ascites)	RT-qPCR on ascites positive.	>3.0y	<10w	1 MU/kg rFeIFN-ω s/c q24h for 36d, reducing to every 3d, followed by 1 × 10^5^ units of rFeIFN-ω *per os* q24h for 2y. PI at 3 mg/kg q48h during 10d. Mirtazapine (Summit Veterinary Pharmaceuticals, Kidlington, UK.) Ursodeoxychloic acid. Cobalamin injections weekly. Itraconazole 10 mg/kg from days 4–87. One Darbepoetin injection. GC-376 s/c Days 17–100. Doxycycline 10 mg/kg bid *per os* from d.32 for 30d (to treat haemotropic mycoplasmosis).	Only for 3d: meloxicam used in preference from Day 4
8	Kitten 2	Effusive (ascites) then non-effusive	FIP Profile.	>2y	<6m	57d of Mutian X (Xraphconn^®^, Mutian Biotechnology Co., Ltd., Nantong, China) at 80 mg/kg. Following her neurological relapse, she was re-treated with 160 mg/kg for 2 m then 1 × 10^5^ units of rFeIFN-ω *per os* q24h.	No
9	Skywise	Non-effusive	RT-qPCR on MLN FNA positive for mutation M1058L, negative for S1060A.	>2.0y	<35d	50d Mutian X starting 160 mg/kg q24h *per os* in divided doses, reduced to 120 mg/kg on Day 25; followed by 1 × 10^5^ units of rFeIFN-ω *per os* q24h. Cobalamin (Cobalaplex).	6d *per os*, then in eye drops for 14d
10	Betsy	Effusive (ascites)	RT-qPCR on ascites C*_T_* 25.6.	>1.5y	<39d	38d Mutian X 80 mg/kg *per os* for 31d and 160 mg/kg for 7d followed by 1 × 10^5^ units of rFeIFN-ω *per os* q24h for 1y. Doxycycline.	5 mg/cat q24h for 10d replaced by meloxicam
11	Dante	Colonic FIP	Histopath of biopsies positive.	>7m	<28d	84d of Mutian X 80 mg/kg *per os*. Protexin pro-kolin enterogenic probiotics 2 mL *per os* q12h. Cobalamin (Cobalaplex).	No: gabapentin 25 mg q8h
12	Elmo	Non-effusive	Biopsy histopath reported pyogranuloma.	>17m	<31d	54d Mutian X (160 mg/kg for 37d, then 80 mg/kg) followed by 1 × 10^5^ units of rFeIFN-ω *per os* q24h.	One injection and one 5 mg pill given once only
13	Lyra	Effusive (ascites)	RT-qPCR on ascites: low positive C*_T_* 33.	>13m	<20d	47d Mutian X 80 mg/kg q24h (divided doses) *per os* followed by 1 × 10^5^ units of rFeIFN-ω *per os* q24h for 5m.	No: meloxicam given instead
14	Molly	Non-effusive	FIP profile. RT-PCR positive feces over 24m.	>3.3y	<16m	1 × 10^5^ units of rFeIFN-ω *per os* q24h, gabapentin, cefovecin (Convenia, Zoetis, Surrey, UK), Synbiotic D-C probiotics (Protexin).	Sliding doses (unspecified) then meloxicam
15	Bea	Effusive (ascites)	RT-PCR on ascites positive. Histopath biopsy positive.	13m † (died of cancer, aged 8y)	<69d	94d of Mutian X: 40 mg/kg (i.e., half dose) for 4d, then 80 mg/kg for 90d, except for one week of double dose, followed by 1 × 10^5^ units of rFeIFN-ω *per os* q24h for 6 m. Robenacoxib (Onsior, Elanco Animal Health, Hampshire, UK) given once only.	No
16	Buddie	Effusive (ascites)	RT-PCR on ascites: C*_T_* 33. Histopath of MLN, spleen, liver biopsies positive.	>14m	<51d	69d of Mutian X 80 mg/kg with 160 mg/kg in the 3rd week of treatment; followed by 1 × 10^5^ units of rFeIFN-ω *per os* q24h for 6m. Cobalamin (Cobalaplex); SAMe (Denamarin).	One dexamethasone injection only
17	Nelson	Non-effusive becoming effusive	IHC of MLN biopsy positive.	>6m	<36d	52d of Mutian X: 7d at 160 mg/kg, reduced to 120 mg/kg then 80 mg/kg; followed by 1 × 10^5^ units of rFeIFN-ω *per os* q24h. Also cobalamin (Cobalaplex); amoxycillin and clavulanic acid (Synulox, Zoetis, Surrey, UK), mirtazapine; cobalamin (Cobalaplex); SAMe (Denamarin).	No: meloxicam instead
18	Wish	Non-effusive	IHC of MLN biopsy positive, RT-PCR positive (negative for mutations).	>12m	<13d but no initial test, so no proof it was ever raised	29d of Mutian X: 7d at 160 mg/kg and 22d at 80 mg/kg; followed by 1 × 10^5^ units of rFeIFN-ω *per os* q24h. Cobalamin (Cobalaplex); Denamarin.	For 7d only
19	Mike	Non-effusive (colonic)	Biopsy histopath positive.	>2.5y	unknown: AGP tested only once after 1 y of treatment	1 × 10^5^ units rFeIFN-ω *per os* q24h for 5m, meloxicam and PI, followed by 12 w of GS-441524 injections (DC Chemicals, USA), followed by 1 × 10^5^ units of rFeIFN-ω *per os* q24h, and PI.	No
20.	Chynah	Effusive (pleural effusion)	RT-PCR on pleural effusion C*_T_* 33.	>7m	remained raised > 3m	rFeIFN-ω 1 MU/kg s/c q48h, reducing to twice weekly for 5m. Repeated drainage of effusion.	2.5 mg/cat q24h sliding doses
21	Tabitha	Non-effusive	FIP profile.	>12m	<29d	58d of Mutian X starting at 160 mg/kg for 8d, then 120 mg/kg for 20d then 80 mg/kg; followed by 1 × 10^5^ units of rFeIFN-ω *per os* q24h. Protexin pro-kolin enterogenic probiotics.	No: meloxicam
22	Harry	Small amount of ascites and enlarged MLN	FCoV RT-PCR positive. Cytology and IHC MLN biopsy inconclusive.	>8m	<30d	50d of Mutian X: 43d at 80 mg/kg then 7d at 160 mg/kg; followed by 1 × 10^5^ units of rFeIFN-ω *per os* q24h. Cobalamin, SAMe.	5.0 mg /cat q12h for 5d
23	Mr Twinkles	Effusive (ascites)	FIP profile.	>8m	<117d	7 mg/kg Spark [53] injection s/c for 10d, followed by 4 Spark 5 mg tablets (i.e., 7 mg/kg) *per os* q24h for 84d. Cobalamin (Cobalaplex). Denamarin.	No
24	Tyra	Non-effusive (uveitis)	FIP profile.	>10m	<68d	Pine and Lucky 9 [53] injections (12–14 mg/kg): Pine for 4d, Lucky injections for 68d. Then 9 Lucky red pills *per os* q24h for 13d.	No
25	Munchie	Effusive (ascites)	FIP profile.	>1.5y	<108d	Spark 8 mg/kg *per os* q24h for 84d.	No
26	Edward	Effusive (pleural effusion)	FIP profile incuding cytology.	10m († RTA)	<7m	rFeIFN-ω 1 MU/kg s/c q48h, meloxicam *per os* q24h.	No

> indicates that survival was over this period † indicates death. AGP—alpha-1 acid glycoprotein. C*_T_*—Cycle threshold. FCoV—feline coronavirus. FNA—fine needle aspirate. GC376—a 3c-like protease inhibitor antiviral drug [49]. GS-441524—a nucleoside analogue antiviral drug [50]. d—days: h—hours: m—months: w—weeks: y—years IHC—immunohistochemistry. MLN—mesenteric lymph node. mRNA—messenger RNA. MU—million units. Mutian X—Mutian Xraphconn^®^ (Mutian Biotechnology Co., Ltd., Nantong, China), an adenosine nucleoside analog [52,65]. PBMC—peripheral blood mononuclear cells. PI—polyprenyl immunostimulant (PI, VetImmune, Sass & Sass, Oak Ridge, USA). Pos—positive. q—indicating how often, i.e., q24h is daily; q12h indicates twice a day, etc. rFeIFN-ω—recombinant feline interferon omega (Virbac, Nice, France). RTA—road traffic accident. RT-PCR—reverse transcriptase polymerase chain reaction. RT-qPCR—reverse transcriptase quantitative polymerase chain reaction. SAMe—S-adenosyl-L-methionine s/c—subcutaneously.

This table shows the clinical presentation of FIP (effusive, non-effusive, colonic, etc.) and what evidence there was for FIP diagnosis (C*_T_* results given when available). Twenty-three of the 26 recovered cats are alive and well at the time of writing; therefore, the over sign (>) precedes their survival duration. The time from the onset of treatment to the first normal AGP test is given in the 6th column. The exact interval from FIP treatment onset to normal AGP level could not be accurately determined because it depended on the frequency of the cat being blood tested. We were only able to determine that it was less than (represented by the < sign) however many days from the start of treatment to the first AGP test WNL. The 7th column lists the treatments given, and the 8th column lists whether corticosteroids were used or not.

**Table 2 viruses-14-00744-t002:** How FIP was diagnosed, survival time, and treatment details of 16 cats who went into remission or died.

	Cat	FIP Presentation	How FIP Diagnosed	Survival in Months	Treatments	Prednisolone/Corticosteroids
1	Yrael	Effusive	RT-qPCR on effusion C*_T_* 32.	1.5 †	Coconut oil and draining the effusion.	Unknown
2	Charlie Chaplin	Non-effusive	RT-qPCR on MLN FNA C*_T_* 25. Histopath positive.	1.5 †	1 × 10^5^ rFeIFN-ω *per os* q24h. Clindamycin for toxoplasmosis co-infection.	✓
3	Smokey	Non-effusive	RT-qPCR on MLN FNA C*_T_* 28. Histopath positive.	1.8 †	Adipose stem cell therapy.	Unknown
4	Rowley	Effusive (ascites), then non-effusive (uveitis, finally severe haemolytic anaemia)	RT-qPCR on ascites C*_T_* 28.4. Histopath positive.	2.25 †	1 MU/kg rFeIFN-ω s/c q48h resulted in resolution of his ascites then 1 × 10^5^ units *per os* q24h. Mirtazapine 3 times. Anti-TNF-alpha monoclonal antibody infliximab, (Remicade^®^, MSD, London, UK) 4 mg/kg in a 0.9% saline infusion over a 4h period. Type B blood transfusion.	✓ 5 mg q12h, dose not reduced
5	Claude	Effusive (pleural effusion initially then also ascites)	Partial FIP profile.	3.0 †	1 MU/kg rFeIFN-ω s/c q48h. Thoracentesis.	✓ 1 mg/kg sliding doses
6	Alfie	Non-effusive (chronic diarrhoea, enlarged MLN)	RT-qPCR on MLN FNA C*_T_* 31, on feces C*_T_* 29. Histopath positive.	3.5 †	rFeIFN-ω given *per os,* but dose not recorded. Fortiflora probiotics (Purina).	✓ dose not recorded
7	Holly	Non-effusive to effusive ascites	mRNA and 3′UTR RT-PCR on ascites positive.	4 †	5 × 10^4^ rFeIFN-ω *per os*. Colloidal silver (dose unknown).	Unknown
8	Bugsy	Non-effusive	FIP Profile.	6 †	1 MU/kg rFeIFN-ω s/c q48h then 1 × 10^5^ units per os q24h.	✓5 mg eod
9	Daisy	Non-effusive	RT-qPCR on MLN FNA C*_T_* 27.	5 †	1 MU/kg rFeIFN-ω s/c q48h then once a week, then 5 × 10^4^ units rFeIFN-ω *per os* q24h.	✓ higher dose reduced to 0.5 mg/kg q24h after 14d
10	Levi	Effusive (ascites)	RT-qPCR on ascites C*_T_* 33.	5.5 †	1 MU/kg rFeIFN-ω s/c twice a week. Ascites drained.	✓dose not recorded
11	Roxanne	Non-effusive and FGS	RT-qPCR on PBMC C*_T_* 42 (very low positive), on faeces C*_T_* 23.	11 †	1 MU/kg rFeIFN-ω s/c twice weekly. Clindamycin (Antirobe, Zoetis, Surrey, UK), meloxicam.	No: meloxicam
12	Pip	Non-effusive	RT-qPCR on PBMC C*_T_* 40 (very low positive) twice.	>12	5 × 10^4^ units rFeIFN-ω *per os* q24h.	✓10 mg/cat pred q24h reducing to 5 mg q24h
13	Pharaoh	Non-effusive	RT-PCR on MLN biopsy C*_T_* 27. MLN biopsy inconclusive.	8 †	Adipose stem cell therapy.	Unknown
14	Maximus	Non-effusive	RT-PCR on MLN FNA positive.	14 †	1 MU /kg rFeIFN-ω s/c q48h	5 mg/cat q24h
15	Ragamuffin	Non-effusive	Biopsy MLN & intestine inconclusive.	>60	1 MU/kg rFeIFN-ω s/c q48h then 1 × 10^5^ units *per os* q24h then 1 MU/kg rFeIFN-ω twice a week	✓Sliding doses
16	Tinkerbell	Non-effusive	FIP Profile. RT-qPCR on feces C*_T_* 31–34.	>36	1 MU/kg rFeIFN-ω s/c q48h then 1 × 10^5^ units *per os* q24h. PI. Mirtazapine 2 mg/cat q48h.	✓For over 1y at 2 mg/kg q24h

✓—present. > indicates that survival was over this period † indicates death. 3′-UTR—three prime untranslated regions of the FCoV gene. AGP—alpha-1 acid glycoprotein. C*_T_*—Cycle threshold. FCoV—feline coronavirus. FGS—feline chronic gingivostomatitis. FNA—fine needle aspirate. h—hours: m—months: y—years. IHC—immunohistochemistry. MLN—mesenteric lymph node. mRNA—messenger RNA. MU—million units. PI—polyprenyl immunostimulant (VetImmune, Sass & Sass, Oak Ridge, USA). PBMC—peripheral blood mononuclear cells. Pos—positive. q—indicating how often, i.e., q24h is daily; q12h indicates twice a day, etc. rFeIFN-ω—recombinant feline interferon omega (Virbac, Nice, France). RT-PCR—reverse transcriptase polymerase chain reaction. RT-qPCR—reverse transcriptase quantitative polymerase chain reaction. SAMe—S-adenosyl-L-methionine. s/c—subcutaneously. This table of 16 cats that only experienced remission shows the clinical presentation of FIP (effusive, non-effusive, colonic, etc.) in the 3rd column; the 4th column gives the evidence for an FIP diagnosis. The 5th column shows the cats’ survival in months: three were lost to follow up. The 6th column lists the treatments given, and the 7th column lists whether corticosteroids were used or not. Their AGP results are shown in Figure 2.

**Table 3 viruses-14-00744-t003:** FIP score before and after treatment in the recovered group.

	Cat	Clinical Signs	FCoV Antibody Titre (Interval First to Last)	Anaemia i.e., Hct <30%	Lymphopenia <1.5 × 10^9^/L	Hyperglobulinaemia i.e., >45 g/L	Score	
							Before	After
1	Basil 1	Icterus	1280/10 (13m)	✓/✗ (22d)	✓/✗	✓/✗ (63–43)	5	0
2	Boris	Poor appetite, diarrhoea	>1280/160 (4m)	✗	✗	✗ (34–40)	2	1
3	Mars	Anorexia	>1280/320 (18m)	✗	✓/✗	✓/✗ (69–42)	4	1
4	Chester	Pleural effusion, pyrexia	>1280/1280 (28m)	✗ *	✗	NA/✗ (NA–43)	2	1
5	Amy	Weight loss, pyrexia, tender abdomen. Icteric plasma.	1280/640 (16m)	✗	✓/✗	✓/✗ (55–38)	4	1
6	Brook	Recurring pyrexia	1280/0 (13m)	✓/✗ (33d)	✓/✗	✓/✓ (57–51)	5	1
7	Basil 2	Ascites	>1280/>1280 (27m)	✓/✗ **	✓/✗	✓/✗ (63–42)	5	1
8	Kitten 2	Ascites. Relapse: painful tail, ataxia, seizures. Full recovery.	>1280/640 (8m)	✗	✗	✓/✗ (58–39)	5	1
9	Skywise	Weight loss, uveitis	>10,240/640 (9m)	✓/✗ (12d)	✓/✗	✓/✗ (91–47)	5	0
10	Betsy	Ascites, profound anaemia (Hct 14%), underweight: BCS 2/9.	>1280/>1280 (2m)	✓/✗ (20d)	✓/✗	✗ (41–44)	4	1
11	Dante	Vomiting and diarrhoea	1280/>1280 (2m)	✗/✗	✗	✓/✓ (68–53)	3	1
12	Elmo	Quiet purr, poor appetite, listless. Pyrexia, mesenteric lymph node enlargement, anaemia.	>10,240/0 (8m)	✓/✗ (31d)	✗	✓/✗ (72–42)	4	0
13	Lyra	Ascites	>10,240/2560 (8m)	✓/✗ (47d)	✓/✗	✓/✗ (69–35)	5	0
14	Molly	Uveitis, chronic diarrhoea: carrier cat shedding virus in faeces >23m	>1280/1280 (15m)	✗	✗/✗ (low)	✓/✗ (57–56)	3	2
15	Bea	Ascites, sudden weakness in limbs. Trichobezoar caused vomiting.	>1280/640 (11m)	✓/✗ (26d)	✗	✓/✓ (78–53)	4	1
16	Buddie	Ascites	>1280/1280 (7m)	✓/✗ (26d)	2 of 7 samples	✓/✓ (106–74)	4	1
17	Nelson	Dull, reduced appetite, enlarged MLN, ascites after biopsy.	640/>1280 (6m)	✗	✗	✓/✗ (79–33)	3	1
18	Wish	Enlarged MLN, chronic diarrhoea	>1280/not re-tested yet	✗	✓/✗	✓/✓ (85–51)	4	2
19	Mike	Chronic diarrhea, haematochezia, continuous virus shedding two years later although cat is well.	>1280/not re-tested	✓/✗ (154d)	✓/✗	✓/✗ (105–45)	5	2
20	Chynah	Cough, pneumonia, nasal discharge, dyspnoea, pleural effusion.	1280/>1280 (8m)	NA	NA	✓/✓ (64–47)	3	2
21	Tabitha	Underweight, pyrexic, watery diarrhoea, ataxic episode, possible uveitis.	>1280/not re-tested	✗ *	✗ (but initial count low at 1.89)	✓/✓ (88–59)	3	2
22	Harry	Small amount of ascites and enlarged MLN	>1280/>1280 (2m)	✓/✗ (30d)	✓/✗	✓/✓ (99–54)	5	2
23	Mr Twinkles	Ascites	>1280/1280 (6m)	✓/✗ (57d)	✓/✗	✓/✓ (58–51)	5	2
24	Tyra	Uveitis	>1280/320 (5m)	✗	✗	✓/✓ (58–54)	3	1
25	Munchie	Ascites, pyrexia, lethargy	>1280/>1280 (4m)	✗	✗	✓/✗ (51–38)	3	1
26	Edward	Pleural effusion	>1280/>1280 (7m)	✗ but 1st count 2m into treatment	✗ but 1st count 2m into treatment	✓/✗ (59–44)	3	1
	**Summary**		**7 of 26 FCoV antibody titres reduced significantly (at least 3-fold)**	**12 of 25 cats anaemic: anaemia resolved in all**	**9 of 24 cats lymphopenic on at least one test**	**22/25 cats had raised globulins, one NA.** **Reduced in all 22 and to WNL in 15 cats.**		

✓—present ✗—absent <—under >—over. Hct—haematocrit. d—days m—months. NA—not available. WNL—within normal limits. * Only one result just under the Hct cut-off of 30% (i.e., 29% on one occasion). ** Number of days not given because this case was complicated by concurrent infectious anaemia.

Table 3 shows the clinical signs before treatment; FCoV antibody titres first and most recent, with the interval in parenthesis; Hct before and after with the time taken for anaemia to resolve in parenthesis; lymphopenia; hyperglobulinaemia, showing a reduction in parenthesis. Anaemia was defined as Hct <30%, lymphopenia as a lymphocyte count under 1.5 × 10^9^ per liter, and hyperglobulinaemia as over 45 g per liter (g/L).

The Summary row gives the totals for how many FCoV antibody titres reduced significantly, how many cats were anemic before and after treatment, how many cats were lymphopenic, and to what extent the globulins reduced. The last two columns are simply an addition of the FIP features in each row before treatment and after being declared recovered.

**Table 4 viruses-14-00744-t004:** FIP profile parameters at FIP diagnosis and at the last available sample in the remission group.

	Cat	Clinical Signs	FCoV Antibody Titre	Anaemia i.e., Hct < 30%	Lymphopenia <1.5 × 10^9^/L	Hyperglobulinaemia i.e., >45 g/L	Score	
							First	Last
1	Yrael	Effusion	>1280/>1280	NA	NA	NA	2	2
2	Charlie Chaplin	Neurological signs. Toxoplasmosis co-infection.	>1280/>1280	NA/✓	NA/✓	✓/✗	insufficient data	
3	Smokey	Failure to gain weight and enlarged MLN.	>1280/>1280	✓/✓	✓/✓	✓/✓	5	5
4	Rowley	Ascites, then uveitis, then haemolytic anaemia, euthanasia.	>1280/640	✓/✓	✓/✓	✓/✗	4	3
5	Claude	Pleural effusion, deteriorated and ascites appeared.	NA	NA	NA	NA	insufficient data	
6	Alfie	Vomiting but bright initially, chronic diarrhoea, enlarged MLN.	>1280/1280	NA	NA	NA	insufficient data	
7	Holly	Persistent pyrexia, weight loss then ascites.	>1280/1280	✗/✗	✗/✗	✓ /✗	3	2
8	Bugsy	Pyrexia, poor body condition, variable appetite.	>1280/>1280	✗/✓	✗/✓	✗/✓	2	5
9	Daisy	Weight loss, intestinal granuloma, raised MLN. Intestinal granuloma resolved with treatment, but cat still died.	>1280/>1280	✓/✓	✓/✓	✓/✓	5	5
10	Levi	Ascites.	>1280/not repeated	✓/✓ (improved though)	✓/✓	✓/✗	5	4
11	Roxanne	Chronic gingivostomatitis, poor condition, poor appetite, jaundice which resolved, but she suddenly developed ataxia and was euthanased.	>1280/320	✓/✓	✓/✓	✓/✓	5	4
12	Pip	Biopsied 2004. February 2005: enlarged MLN, weight loss, anaemia, reported well in February 2006 although pyrexic (39.2 °C).	>1280/>1280	✓/✓	low/✗	✓/✓	5	4
13	Pharaoh	Weight loss, gut biopsy showed inflammation.	640/>1280	✓/✓	✗/✓	✗/✓	3	5
14	Maximus	Chronic diarrhoea, poor appetite. Collapsed and was euthanased.	>1280/>1280	✓/✓	✗/✗	✓/✓	4	4
15	Ragamuffin	Weight loss. MLN enlarged, diarrhoea, always shed low amounts of virus in faeces, chronic anaemia.	>1280/>1280	✗/✓	✓/✓ (sometimes low normal)	✓/✓	4	5
16	Tinkerbell	Underweight, chronic poor appetite.	>1280/1280	✓/✗	✗/✗	✓/✓	4	3
	**Summary**		**15 of 15/** **13 of 14**	**9 of 12/** **11 of 13**	**6 of 11/** **10 of 13**	**11 of 13/** **9 of 13**		

✓—present ✗—absent >—over. NA: not available.

Table 4 shows the clinical signs and scores before treatment and at death or last sampling prior to death or loss to follow up, showing that treatment had not greatly improved the cat’s score, and in some cases it worsened. First and latest FCoV antibody titres, presence or absence of anaemia, lymphopenia, and hyperglobulinaemia are shown in the 4th, 5th, 6th, and 7th columns, respectively. In the last row, a summary of how many cats were anemic, lymphopenic, and hyperglobulinaemic at the first and last samples are shown. FCoV antibody titres were available for 15 of 16 cats, and all were high. 13 of 14 last available samples had high antibody titres, one cat’s titre had reduced three-fold from >1280 to 320; 9 of 12 cats were anemic at the outset, and at the last sample, 11 of 13 were anemic. Six of 11 cats were lymphopenic, rising to 10 of 13; 11 of 13 cats were hyperglobulinaemic at the outset, globulin levels reduced in four cats, probably due to corticosteroids. FIP scores worsened in 3 cats, stayed the same in 4 cats, and improved slightly in 6 cats. The last two columns are simply an addition of the FIP features in the row at the time of FIP diagnosis and the last available sample prior to death or loss to follow-up.

### 3.2. FIP Treatment

Treatment protocols were variable: treatment choice was at the discretion of the cat’s attending veterinary surgeon or, more recently, the cat’s guardian, and treatments are described to the best of our ability for each cat in Table 1 and Table 2. They most commonly included recombinant feline interferon omega (rFeIFN-ω, Virbac, France) by subcutaneous injection (for effusive FIP) or diluted and administered *per os* daily (for non-effusive FIP or as follow up to other therapies) (*n* = 20 recovered and 13 remission cats). Twelve cats were treated with a 5% oral adenosine nucleoside analogue (Mutian Xraphconn^®^, Mutian Biotechnology Co., Ltd., Nantong, China) [52,65], now known to contain GS-441524 [52], and three cats were treated with online products of unknown active ingredient but suspected to contain GS-441524.

Supportive treatment included vitamin B12 (cobalamin) either as weekly injections or daily pills (Cobalaplex, Protexin Veterinary, Somerset, UK); and Pro-kolin enterogenic probiotics (Protexin Veterinary, Somerset, UK); prednisolone (*n* = 11/25 recovered and 11/12 remission cats for whom use of corticosteroids was known). The use of corticosteroids was recommended to be discontinued or not started by one author (DDA) from 2017, and to use meloxicam *per os* daily instead (after a suitable wash out period and provided blood pressure and kidney function were normal). 

### 3.3. Recovery, Remission, and Death

Case 1 (Basil 1) set the benchmark for defining recovery from FIP against which other cases were compared. This cat was treated and was followed up for eleven years without relapsing and died of chronic kidney disease at 15 years of age. Based on this case, the criteria for declaring a cat recovered from FIP included the following:The cat returned to clinical normality, specifically resolution of the clinical signs of FIP.Globulin levels were reduced to normal (≤45 g/L) or at least significantly reduced (often by over 15 g/L as shown in Table 3).Resolution of lymphopenia, where present, (normal defined as ≥1.5 × 10^9^/L).Haematocrit level increased to normal (defined as ≥30%), with reversal of non-regenerative anaemia where present.At least three-fold reduction in FCoV antibody titre.AGP levels returned to normal levels (≤500 μg/mL) [40].

Cats were deemed to be in remission if only some of the criteria listed above were met: for example, remission cat 15 (Ragamuffin) survived over 5 years, but she was never clinically well during that time, required continuous treatment, and unfortunately, she was lost to follow up.

As shown in Table 1 and Table 2, 26 cats recovered from FIP, with follow-up periods of up to 11 years. Death due to a non-FIP reason occurred in three cats who had recovered from FIP: Basil 1, Bea, and Edward died of chronic kidney disease, cancer, and a road accident aged 15, 8, and 3 years, respectively.

Of the recovered cases, 12/26 (46%) had effusive FIP, and 14/26 (54%) were non-effusive. One cat (Boris) had non-effusive FIP, initially thought to be effusive, but later established to be a cardiogenic effusion (FCoV RT-PCR on his effusion was negative). One cat (Nelson) with non-effusive FIP developed an effusion following biopsy. Two of the non-effusive FIP cases were colonic presentations.

Sixteen cats experienced remissions of 1.5 to over 60 months. Median remission was 4.5 months (not including the outlier of Ragamuffin, who was lost to follow up after 5 years because her result would skew the median). Thirteen cats died or were euthanased, and three were lost to follow up.

Amongst the cats who experienced remission, four (25%) had effusive FIP (one—Rowley—became non-effusive), and 12 (75%) had non-effusive FIP (one—Holly—became effusive).

Twelve of 16 (75%) effusive cases and 14/26 (54%) non-effusive FIP cases recovered. The initial effusive or non-effusive form did not affect whether or not a cat fully recovered (Fisher’s exact, *p* = 0.21).

### 3.4. AGP Levels

AGP levels of recovered cats and those who experienced remission are shown in Figure 1a–c and Figure 2 respectively. AGP levels were elevated (i.e., above 500 μg/mL) in all FIP cases except in the cases of Wish and Mike, where the first AGP test was taken after treatment had started, so the pre-treatment AGP results were unknown. The median AGP level in effusive FIP was 4330 μg/mL, and for non-effusive FIP, 2600 μg/mL. Five cats with non-effusive FIP had levels below 1500 μg/mL.

The time for AGP levels to return to normal in recovered cats varied from a minimum of less than 13, 20, and 22 days (Wish, Lyra, and Basil 1) to over 16 months from onset of treatment, possibly depending on the treatment being used (Table 1). The AGP levels of one recovered cat (Chynah) did not return to normal, and with the benefit of hindsight, this cat should perhaps have been in the remission group, although her veterinary surgeon reported that she was clinically recovered, and unfortunately, she was lost to follow up.

Figure 2 shows the AGP levels of cats in remission and illustrates that cats who did not fully recover from FIP retained high AGP levels: in Ragamuffin’s case, very high levels. The AGP levels in these cats were often suppressed by prednisolone treatment. 

Two of the cats (Holly and Daisy) who did not recover had AGP levels that were reduced to under 500 μg/mL on one occasion each, and AGP levels were reduced to under 1000 μg/mL in two more cats. Therefore, it is important that two consecutive normal AGP results at least one week apart be obtained to ensure that recovery from FIP has occurred.

Of particular interest was the AGP level of Kitten 2 (Figure 1c), who was diagnosed with effusive FIP in October 2019. Her guardian stopped the oral adenosine nucleoside analogue (Mutian Xraphconn) treatment on 5 January 2020, following an AGP result of 709 μg/mL. This young cat presented with an FIP relapse in February 2020, manifesting as hyperesthesia of the tail, which progressed to ataxia, then seizures. A second course was administered, this time at a double dose (one Mutian X 200 pill per kg) which is a level that enables sufficient antiviral drug to cross the blood-brain barrier (Tony Xue, Mutian CEO, personal communication). The cat improved within 24 h, and the second course of Mutian X was continued until AGP reached below 500 μg/mL: she is alive and well two years later. After this event, FIP cat guardians were recommended to give a double dose of oral (not injectable) Mutian (i.e., Mutian X, not Mutian II) for 7–10 days to clear the virus from the brain to prevent neurological relapse.

**Figure 1 viruses-14-00744-f001:**
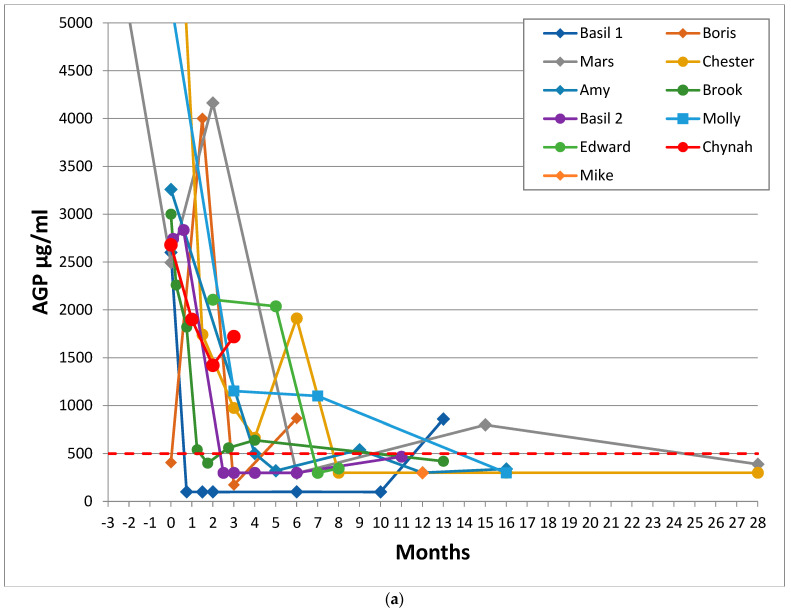
Sequential AGP results of 26 cats who recovered from FIP. These graphs show the time in months for AGP levels to return to normal for the cured cats, with Day 0 being the first day of a specific (as opposed to supportive) FIP treatment (where known: if unknown then Day 0 was the first AGP test). (**a**) shows the AGP levels of 11 recovered cats treated before adenosine nucleoside analog antiviral drugs were available (but including Mike, for whom GS-441524 injections became available 8.5 months after he began treatment with oral of rFeIFN-ω and meloxicam), (**b**) shows the AGP levels of 14 recovered cats treated with products believed to be adenosine nucleoside analog antiviral drugs and (**c**) shows the AGP level and timeline of the 26th recovered cat, Kitten 2, who suffered a neurological FIP relapse when treatment was stopped before her AGP level had fully returned to less than 500 μg/mL; she was re-treated and made a full recovery. The red dashed line indicates the normal AGP cut-off at 500 μg/mL. The *y* axis of (a) and (b) was cut-off at 5000 to facilitate reading of the graphs. Cats with effusive FIP are represented by circle markers, cats with non-effusive FIP are shown in diamonds/squares. No cat had more than one AGP test before treatment began, which is why it appears as if AGP was decreasing prior to treatment—in reality, it would have been increasing. These graphs show that the AGP levels of recovered cats reduced to 500 μg/mL or below, whereas the AGP levels of cats in remission shown in Figure 2 stayed above that level.

**Figure 2 viruses-14-00744-f002:**
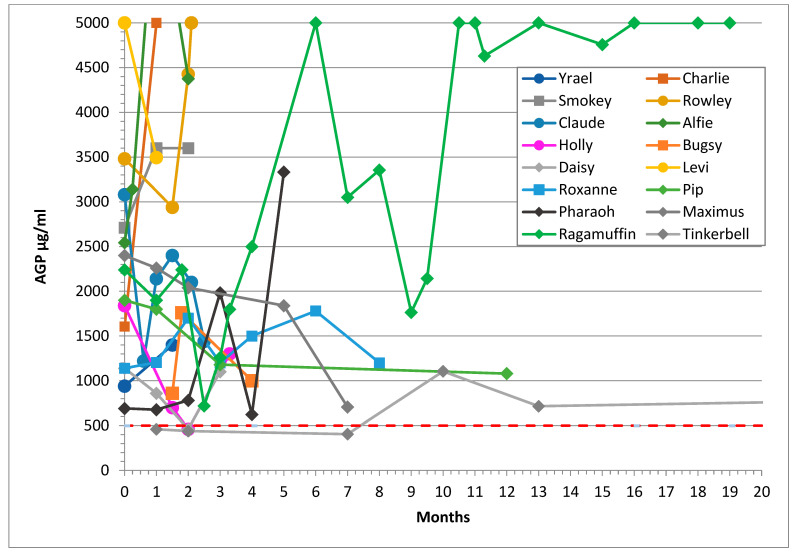
Sequential AGP results of 16 cats who experienced remission from FIP showing that their AGP levels did not reduce to less than 500 μg/mL (shown as the dashed red line) despite treatment. The *y* axis was cut-off at 5000 to facilitate reading of the graphs. Their lines terminate at the last AGP level before death except for the three cats lost to follow up. Median survival time was 4.5 months.

### 3.5. Haematocrit

A lower cut-off of 30% was our definition of normal, below which a cat was said to be anemic. The value was determined by the University of Glasgow Veterinary Diagnostic Services (VDS) Laboratory.

Haematocrit was available for 25 of 26 recovered cats and 13 of 16 cats who went into remission (Table 3 and Table 4). One recovered cat (Chynah) and 3 remission cats (Yrael, Claude, and Alfie) had no haematology results. In five recovered cats (Dante, Molly, Nelson, Edward, Kitten 2), records were not available until after treatment began—too late to determine whether or not they had been anemic because by that time they were not anemic. In two cats, Hct was only marginally below normal (Tabitha 29.4% and Chester: 28.9%) on one occasion only; therefore, they were not seriously anemic and were discounted from analyses. Basil 2′s anaemia was complicated by concurrent haemoplasmosis infection (Figure 3). This left usable hematocrit records for 17 recovered cats and 13 cats who went into remission: 7 of the 17 recovered cats had effusive FIP, and 10 had non-effusive FIP; 2 remission cats had effusive FIP, and 11 remission cats had non-effusive FIP.

Eight of 9 (89%) cats with effusive FIP and 15 of 21 (71%) cats with non-effusive FIP were anemic (Table 3, Table 4 and Table 5). Whether FIP was effusive or non-effusive made no statistical difference to the likelihood of the cat being anemic (*p* = 0.70).

Eleven of 17 (65%) cured cats and 12 of 13 (92%) cats in remission were anemic: the presence of anaemia did not affect the cat’s chances of recovery (*p* = 0.10).

Two remission cats became anaemic after FIP diagnosis, and one cat’s anaemia resolved. Anaemia resolved in all of the cats who recovered, but Basil 2 required a one-month course of doxycycline for concurrent haemotropic mycoplasma infection for his anaemia to resolve (Figure 3).

While the average time for resolution of purely FIP related anaemia was 30.4 days in 10 cats (range 12 to 57 days: Table 3), from initiation of treatment, the average time for AGP to return to WNL was 45 days (range 20–117 days, not counting the two cats for whom there was no pre-treatment AGP). The outlier cat, Mike, was not counted in this analysis because there was no initial AGP for comparison: his anaemia resolved 154 days after treatment began.

### 3.6. Lymphopenia

A lymphocyte count of 1.5 × 10^9^/L was considered the lowest level at which lymphocyte count could be considered normal: this is the cut-off set by the VDS laboratory. Sequential lymphocyte counts were available for 25 of 26 recovered cats and 13 of 16 remission cats, but the first count was two months after treatment began for one recovered cat (Edward). Sixteen (43%) of the remaining 37 cats with FIP were lymphopenic on the first available lymphocyte count closest to FIP diagnosis. The prevalence of lymphopenia was 38.5% (5/13) in effusive cases and 45.8% (11/24) in non-effusive cases.

It appeared that fewer recovered cats (9/24: 37%) than remission cats (7/13: 54%) were lymphopenic, but the difference was not statistically different (*p* = 0.5). Although the median lymphocyte count was slightly higher for recovered cats (1.833 vs. 1.366 × 10^9^/L), the difference was not significantly different between the recovered or remission groups (*p* = 0.47).

Four recovered cats had very low lymphocyte counts but within the normal range (1.5 to 7.0 × 10^9^/L): one cat (Buddie) was lymphopenic on 2 of 7 samples, and another cat (Molly) had low lymphocyte counts (below 2 × 10^9^/L, but above 1.5 × 10^9^/L which is considered the lower cut-off).

Lymphopenia resolved in all recovered cats. In the remission group, three cats became lymphopenic, one cat’s lymphocyte count increased to 1.9 × 10^9^/L, and the status of the other cats remained the same (Table 4).

### 3.7. Hyperglobulinaemia

Globulin levels were available for all the 26 recovered cats and for 13 of 16 remission cats, but there was no pre-treatment sample for one of the recovered cats, and by the time he was tested, his globulin levels were normal. Twenty-three of the remaining 25 recovered cats were hyperglobulinemic: globulin levels reduced in all 23 cats, to WNL in 16 cats, but were still elevated in 7 cats.

Eleven of 13 remission cats were hyperglobulinemic, and globulin levels were reduced in 4 cats.

AGP was a more accurate prognostic indicator than globulin reduction because globulins were slower to reduce than AGP levels in recovered cats (7 of 23 cats still had elevated globulin levels once recovered), and they reduced in 4 of 11 remission cases.

### 3.8. FCoV Antibody Titre

As shown in Table 3 and Table 4, all of the cats had very high FCoV antibody titres, being on or above the upper cut-off point for the laboratory to which samples had been sent. Two exceptions were Nelson, who recovered, and the remission cat Pharaoh, whose FCoV antibody titre was only moderately high at 640 at the time of diagnosis, but subsequently became very high. FCoV antibody titres remained high in almost all recovered cats for a very long period, even years (Table 3). The earliest that a significant reduction in FCoV antibody titre was seen was at 4 months post-diagnosis, and one cat became seronegative at 8 months.

FCoV antibody titres in the remission group did not decrease, except for Rowley, whose titre decreased from >1280 to 640, likely due to immunosuppressive amounts of prednisolone with which he was being treated.

## 4. Discussion

Recovery implies that the disease is finished once and for all, whereas remission is defined as the reduction or diminution of clinical signs, implying that the disease could reappear, i.e., relapse. The state of remission from FIP in our series usually culminated in death, mainly by euthanasia due to a prolonged cachexic state, or sometimes acutely, presenting as acute haemolytic anaemia, collapse, or the onset of neurological signs. Neurological relapses were a problem in previous studies [49,50,51]. While parameters and a scoring system for predicting imminent death have been well documented [34], no such system has been previously defined to differentiate recovery from remission. We present the first documentation of sequential AGP testing of cats being treated for FIP and present evidence that a consistent reduction in AGP to normal levels is the most useful marker for differentiating recovery from remission, and is a clear indicator that it is safe to stop administering adenosine nucleoside analogue antiviral drugs.

Raised AGP was shown to be 100% and 93% sensitive in FIP diagnosis [40,43,44] but is not specific: it rises in both transient FCoV infection [41,42] and in other infections [37,38] (e.g., bacterial peritonitis or pleurisy; *Mycoplasma*
*haemofelis* infection [39]). However, the sensitivity of AGP measurement in cats with FIP in the brain only has not been established: Rissi et al., 2018 [66] reported that five of 22 cats with neurological FIP had no FIP lesions in other organs, and it is possible that in those cats AGP measurement would not have been raised. Raised AGP was more sensitive than histopathology for diagnosing FIP in challenging cases [43], and this was also true in our series, where histopathology—especially of the MLN—was frequently non-specific and reported as simply pyogranulomatous inflammation.

AGP was the most useful parameter for assessing recovery from FIP because it was raised in all our cases, whereas the resolution of anaemia, lymphopenia, and hyperglobulinaemia were not consistently found. Anaemia resolved more quickly than did AGP in five of six of the nine anaemic cats in which the time to return to normal was not the same for both AGP and Hct (in three cats, the identical intervals presumably reflected the intervals between blood samplings). Consequently, where AGP measurement is not possible and if the cat is anaemic purely due to FIP (e.g., uncomplicated by concurrent haemoplasmosis), then the return of Hct to over 30% appears to be a good indicator of recovery, although our numbers were small. However, the anaemia of one of the cats in the remission group also resolved, so the resolution of anaemia alone is not a guarantee of recovery.

The exact interval from the start of FIP treatment to normal AGP could not be accurately determined because it depended on the frequency of the cat being blood tested, which was at the discretion of the cat’s guardian. We were only able to determine that it was less than however many days from the start of treatment to the first AGP test WNL. However, the intervals appeared noticeably shorter after 2019 (Figure 1b vs. Figure 1a), when adenosine nucleoside analogue antivirals were introduced [50]. The earliest that AGP returned to normal was around two to three weeks from the start of treatment. Another acute phase protein, SAA, was also shown to reduce rapidly in cured cats after the onset of Mutian X treatment in a study of 18 cats [52]. 

The prevalence of lymphopenia in our study (43%) was similar to that of other studies: 49.5% of 184 cats with FIP [22], 50% of 106 cats [35], and 64% of 45 cats [34]. In the study by Riemer et al. 2016 [22], lymphopenia was observed significantly more often in 139 cats with effusion and was documented in only 26.8% of 41 cats without effusion. Our results differed: 38.5% of effusive cases were lymphopenic compared with 48.8% of non-effusive cases, but our cohort of cats was much smaller than that of Riemer et al., 2016 [22].

In the study of naturally infected cats by Tsai et al., 2011, the prevalence of lymphopenia was 64% at initial presentation and increased to 91.7% from zero to three days before death [34]. In an experimental infection of 20 cats, the absolute lymphocyte count in blood was the most accurate predictor of disease outcome [36]; therefore, it was surprising that in our study absence of lymphopenia was not a predictor of which cat would recover. However, in our study, lymphopenia reversed in the lymphopenic cats who recovered, and two cats in the remission group became lymphopenic. Accordingly, the development of lymphopenia in a cat with FIP who had not previously been lymphopenic is a poor prognostic sign. Our study could not assess the utility of documenting the reversal of lymphopenia as a marker for recovery from FIP, because unfortunately during most of this study prednisolone was given to treat FIP, which would have interfered with (i.e., decreased or suppressed) the lymphocyte count. Further studies will be required to assess the utility of lymphopenia reversal as an indicator of FIP recovery, although given that around half of cats with FIP are not lymphopenic, it will clearly be less useful than AGP, which is consistently elevated in cats with FIP.

The definitive marker for recovery from FCoV infection is the reduction of FCoV antibody titre to undetectable levels, signifying that viral antigen no longer remains in the body to stimulate an immune response; but this takes many months to achieve and consequently it is not a useful marker to determine when to stop nucleoside analogue treatment (though we used falling FCoV antibody titres to determine when to stop the oral interferon omega follow up). In FIP recovered cats, significant reduction of FCoV antibody titre takes so long to achieve (usually over one year) that it would be inadvisable to sustain nucleoside analogue antiviral treatment long term due to the risk of toxic adverse effects to the kidneys and liver. In contrast, antibody levels in FCoV infected cats without FIP usually decline within a few months of the virus being eliminated from the intestine and faeces (data not shown).

We observed two relapsed cases who presented with a painful tail: Kitten 2, described in this paper, and another cat for whom we had no AGP results and was not documented here. The occurrence of idiopathic painful tail syndrome has been reported previously [67], and André et al., 2019 [68] described a neurological FIP case that began as plegia of the tail. FIP is the major cause of hydrocephalus in young cats [27,28,66]. Cerebrospinal fluid (CSF) drains from the thecal sac at the level of the second sacral vertebra. It is likely that the build-up of CSF in this area pressed on the nerves of the cauda equina, causing an initial presentation of sensitivity and pain in the tail. As CSF built up in the ventricular system of the central nervous system, the two cats progressed to show hind limb ataxia, then seizures. Fortunately, prompt administration of antiviral pills to Kitten 2 saved her life; the other cat was treated with an injectable nucleoside analogue (Mutian II) but died. Two other cases (outwith this study) treated with injectable nucleoside analogues have presented with neurological relapses, but made partial recoveries after a course of double dose pills: it seems counterintuitive for an oral formulation to cross the blood-brain barrier more effectively than the injectable formulations, but that is our experience. Following the Kitten 2 relapse, we recommend giving a 7–10-day double dose of oral GS-441524 early in the treatment to clear the brain of the virus. 

It might be wondered why there was only one relapse amongst the 16 cats (6%) which were treated with nucleoside analogue drugs when in the first study where nucleoside analogues were used, there were eight relapses in 26 cats (31%) [50]. We believe that one of the most important reasons was that 12 cats were treated with an oral adenosine nucleoside analogue, which accesses the site of viral replication in the intestine and stopped virus shedding in faeces in all cases, rather than an injectable version of the same drug. In contrast, one cat (Mike) treated with injectable GS-441524 continued shedding virus two years after diagnosis. A similar problem has been encountered in humans treated with Remdesivir (Gilead Sciences, Foster City, CA, USA), the GS-441524 pro-drug (with or without convalescent plasma), where prolonged virus shedding was reported in 13 patients [69]. It may be that this antiviral fails to adequately reach every organ, allowing pockets of the virus to survive and recrudesce [70]. One patient not in our series (because AGP was not monitored) was presented with a relapse of FIP pleural effusion. Initially, that cat had been treated with ten days of Mutian II injections which may have failed to properly clear the virus from his lungs: this was seen in an immunocompromised human COVID19 patient treated with Remdesivir [70].

Other factors which contributed to a successful outcome included being able to differentiate accurately between recovery and remission using AGP levels, therefore knowing when it was safe to discontinue the antiviral drug. Secondly, antiviral treatment was followed up with long-term oral feline interferon, which has antiviral and immunomodulatory activity. Thirdly, in-contact cats were tested for FCoV shedding, and if positive, they were treated to stop virus shedding [65]; thus, re-infection was prevented (presumably, re-infection could appear as a relapse). Using antivirals in subclinically infected cats parallels SARS-CoV2 infection, where prophylactic ivermectin reduced the risk of COVID-19 disease by 83% in the following month in health care workers in India [71] and reduced COVID-19 mortality in Brazil by 68% [72].

It should always be considered that the appearance of relapse or failure to respond to treatment may, in reality, be due to secondary conditions. For example, the anaemia of Basil 2 was due to concurrent haemoplasmosis but initially appeared to be a lack of response to FIP treatment.

It is likely that immunosuppression due to FIP-induced lymphopenia leads to secondary infections in FIP. Thus, in addition to infectious anaemia, we have observed toxoplasmosis and clinical signs attributable to recrudescent latent feline herpesvirus infection, such as epiphora and sneezing. One relatively frequent curious condition was trichobezoar which is not explicable by immunosuppression. In most cases, this was easily dealt with using a psyllium-containing dry cat food (Aging 12+, Royal Canin, Aimargues, France), but in one cat, who was not included in this series due to lack of AGP results, surgery was required to remove a large furball.

A failing of our study was the inability to compare weight gain or loss in the two groups. In a previous study, weight gain was an important measurement of treatment success [50]. Unfortunately, in our study, weight records were only available for the most recent recovered cases and none in the remission group.

Another limitation of the study was the small number of cats involved. However, our figures are in the same ballpark as other recently published FIP treatment studies, which featured 20 [49], 18 [52], and 31 [50] FIP cats, respectively. Since the advent of effective treatments for FIP, it is hoped that our control group of remission cats will not be increased.

The data presented here were collected over almost two decades. As new tests became available, they were applied to the cases; consequently, confirmation of FIP diagnosis of individual cats was not uniform: obtaining reliable and complete data is one of the drawbacks of a retrospective study such as ours. Demonstration of FCoV in biopsy or post mortem material by immunohistochemistry is deemed the gold standard for FIP diagnosis, and this study might be criticised for a lack of histopathological confirmation of FIP in many of the cases. Since the cats were field cases, it would not have been ethical to ask for biopsies to be performed for this study, and most guardians did not elect to have histopathological confirmation of the diagnosis in those cats who died. Many of the cats had positive FCoV RT-PCR tests of effusion [54,55,56] or MLN FNA [57], and such evidence has been accepted as diagnostic of FIP in other therapeutic studies [49,50]. Demonstration of replicating FCoV by messenger RNA RT-PCR [58] in PBMC was considered diagnostic of FIP, but it was borne in mind that FCoV mRNA had been detected in 5% of cats without FIP [58,59], and during this study, we found that the primers cross-reacted with human DNA, giving false-positive results on some samples (therefore we confirmed positive PBMC mRNA results by a 3′ UTR RT-PCR [60]).

There remained some cats who were diagnosed only by circumstantial evidence, but in all cases, elevated FCoV antibody titres and AGP levels were documented, and response to anti-coronavirus drugs was further corroboration of a correct FIP diagnosis.

## 5. Conclusions

As new FIP treatments become available, we have shown that AGP measurement will be a useful parameter to assess their efficacy. Increasing AGP can indicate that a cat’s condition is deteriorating, but a return to 500 μg/mL or less sustained for at least one week indicated recovery from FIP and differentiated recovered cats from those in remission. It is inadvisable to reduce the dosage or frequency of treatment, even if clinical signs appear to have resolved, until AGP levels are once again normal.

## Figures and Tables

**Figure 3 viruses-14-00744-f003:**
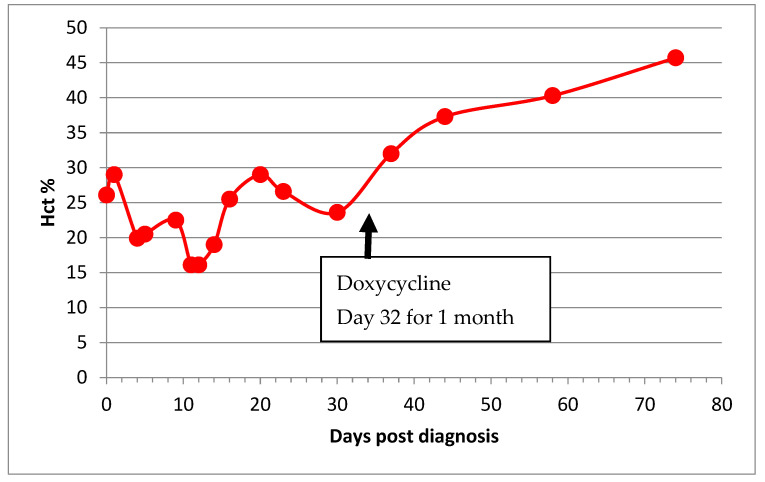
Haematocrit (Hct) of a cat (Basil 2) with both FIP and haemotropic mycoplasmosis illustrating the typical wave pattern with a periodicity of 7–10 days due to the cyclical parasitemia of haemotropic mycoplasma infection. The co-infection compounded the effect of the anaemia of FIP, doxycycline was introduced on Day 32, and his anaemia normalised thereafter.

**Table 5 viruses-14-00744-t005:** Summary of anaemia.

Title 1	Anaemic	Not Anaemic	Total
Recovered effusive	6	1	7
Recovered non-effusive	5	5	10
Remission effusive (but 1 became non-effusive)	2	0	2
Remission non-effusive (but one became effusive)	10	1	11

## Data Availability

All data generated or analysed during this study are included in this published article.

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
