# Peer review of "Alpha-1 Acid Glycoprotein Reduction Differentiated Recovery from Remission in a Small Cohort of Cats Treated for Feline Infectious Peritonitis"

_viruses, 2022, doi:10.3390/v14040744_

Round 1

Reviewer 1 Report

In this article, the authors evaluated data from 42 cats diagnosed with feline infectious peritonitis (FIP) over 21 years. The main objective is to find a parameter or marker that can be used to differentiate recovery from remission in infected cats and thus help clinicians decide when to cease antiviral treatment. The authors propose that the acute phase protein alpha-1 acid glycoprotein (AGP) is a useful parameter in differentiating recovered cats from remission because elevated AGP levels in cats with FIP return to normal values ​​more quickly and consistently than other biochemical or immunological parameters in recovered cats.

In general, it is an interesting manuscript although there are data and text that must be revised.

The title is too categorical considering that the data comes from 42 cats diagnosed with FIP, of which 26 are recovered cats and 16 are cats in remission or dead. This is a low number of animals compared to other retrospective studies such as Riemer et al 2016, who reviewed data from 231 confirmed FIP cats.

Abstract: the authors must verify the numbers of cats included in the results of anaemia and lymphopenia and include that these results are not significantly different between recovered or remission groups. The total number of cats included in the study should be mentioned, with the percentages of effusive and non-effusive cases in recovered and in remission cats.

INTRODUCTION:  the authors should include:

  • Description of the mutation of FoCV strains to develop FIP strains, including the mutations of interest and their relation to systemic spread of the virus (as it is mentioned in section 2.2. and in tables 1 and 2)
  • A better description of FIP pathogenesis, the effusive and non-effusive forms and their possible outcomes, recovery vs. remission
  • A better description of FIP diagnosis (included in Material and Methods 2.2) and what is a FIP profile. The authors mention that the cats were selected if sequential laboratory tests were available, but it is not clear if they have followed a diagnostic protocol such as that recommended by the ABCD in 2021
  • In the case of AGP, it is important to clearly indicate the values ​​that are considered normal, the high ones that may be suspicious for FIP, and the very high ones that could support the diagnosis of FIP (see Tasker 2018), keeping in mind that moderate AGP elevation per se are not specific for FIP. This information must also be included in material and methods and results sections. And the AGP value included in the Conclusions section must be corrected.

MATERIAL AND METHODS:

In general, much of the information in this section should go into the results or even discussion section. In Material and methods, only the approach of the study should be explained: how the cats were selected, how the FIP diagnosis was made and the FIP profile, the parameters and data that are collected, the laboratory techniques used, and the statistical analysis used. Results and their interpretation or discussion should not be included (e.g. Lines 88-97).

The cats must be identified by an alphanumeric code, not by their name (cat 1, cat 2, etc).  It is easiest to identify the cat by the number (Table 1: cat 1 to cat 26 are recovered; Table 2: cat 27- 42 are remission or die). In addition, this type of identification helps to understand the results obtained and the discussion.

Tables 1, 2, 3 and 4 are actually results, and I suggest moving them to the Results section. Please check data and figure legends (some abbreviations are missing).

In Table 1 and 2: Identify cats with histopathology studies and their positive results. Write the RT-qPCR and Ct results in the same way. Type PMBC instead of pbmc. Is survival calculated in months or years from the diagnosis of FIP? Has this data been obtained in the same way in all cases?

Table 3 and 4: the interpretation of the results (“Total”) should not be included in the tables. “Before” and “after” are lost in the score column of Table 4. Normal values ​​or haematocrit, lymphocytes, and gammaglobulins should be mentioned somewhere. What techniques were used to detect FoCV antibody titers? How were the scores calculated before and after treatment?

The interpretation of the data from the cats (RT-qPCR, anemia, AGP, etc.), whether they fulfilled multiple diagnostic criteria for FIP, or the usefulness of AGP instead of histopathology for diagnosis (lines 155-166) and the evolution of clinical signs, scores, etc. in tables 3 and 4 (lines 179-186) should be moved to the Results and/or discussion section.

Authors should review the Treatment (section 2.3) and Results (section 2.4) as much of the description should be in the results and/or discussion sections (eg, the outcomes of three cats or the criteria for declaring that a cat has recovered from FIP infections). The description of the possible outcomes also fits better in the introduction of the manuscript.

RESULTS AND DISCUSION:

3.1. Please, check the number of cats that died or were euthanized and the number lost to follow up.

Figures 1 and 2: It is very difficult to interpret the figures of AGP results and it is impossible to see when the AGP values ​​return to normal in each cat. Perhaps the authors could try a different type of graph or include fewer values ​​for each cat. Why only 25 cats in figure 1?

Line 297: The idea that cats in remission would be the equivalent of long-COVID in humans is interesting, but not supported by the results of this study in a very small number of cats (16). In any case, the authors could only state that it is an assumption since the mechanisms of long-COVID are not fully understood, and therefore I suggest removing it.

Line 303-313. The authors must explain if the cat in Figure 3 is in the recovery or remission group. I suggest writing "cat owners" instead of "cat guardians". In addition, check the data in Table 5 and its interpretation in section 3.3, it is difficult to understand.

The text explaining figure 4 is repeated before and after that figure (lines 346-348 and 352-355)

3.4. The authors must explain why only 37 cats with PIF are included in the lymphocyte count instead the 42 cats in the study.

LINE 405: Authors should mention PIF when describing recovery and remission in the first paragraph.

In the critical aspects of the study, the authors should also mention the low number of animals included in each group (26 recovered, 16 in remission) and also, the difficulty of obtaining reliable and complete data as it is a retrospective study.

Reviewer 2 Report

Feline infectious peritonitis is a systemic immune-mediated perivasculitis usually fatal which occurs in a minority of FCoV infected cats. This is an interesting study on the evaluation of the acute phase protein alpha-1 acid glycoprotein (AGP) as parameter to differentiate FIP recovery from remission.

Some observations:

Table 3: In columns 5-6-7 it is not specified for all cats the number of days before and after the treatment. Moreover, why only for cats 6, 11, 15,16, 1821, 22, 24, 25 and 26 are reported the values of Hypergamma globulinemia (before and after treatment)? Only cat 1 has the values of anemia before and after treatment. And the values of other cats?

Line 174: “This table”: please specify it is table 4.

Line 174 and table 4. There is incongruity. Table 4 reports data before and after treatment

Table 4: It is not clear the last line “total”

Lines 230-232: it is not clear. Please rewrite and clarify the test

Figures 1 and 2 are difficult to interpret. There are too many data/lines. They should be redone making them more readable

Lines 319-320: Review the formatting

Author Response

Addie et al: To both reviewers: our changes in the manuscript are highlighted in yellow for ease of viewing (except the figures which I couldn’t make yellow).  The semi-final version shows you what parts were deleted or moved, the final version is the cleanest version, where, if accepted, I’ll only need to remove the yellow highlighting of the changes.

We want to thank both reviewers for your time and for making our paper a much stronger one.

Reviewer 2. Feline infectious peritonitis is a systemic immune-mediated perivasculitis usually fatal which occurs in a minority of FCoV infected cats. This is an interesting study on the evaluation of the acute phase protein alpha-1 acid glycoprotein (AGP) as parameter to differentiate FIP recovery from remission.

Addie et al: We want to thank Reviewer 2 for reading our paper, for your favourable review, for the improvements you have kindly suggested and the time you clearly spent reading the manuscript in detail, and for your rapid turnaround time.

Reviewer 2. Some observations:

Table 3: In columns 5-6-7 it is not specified for all cats the number of days before and after the treatment. Only cat 1 has the values of anemia before and after treatment. And the values of other cats?

Addie et al: In column 5 (anaemia) the number of days to resolution of anaemia was only missing for Basil 2 and that was because his case was complicated by concurrent infectious anaemia.  I have now inserted 2 asterisks and an explanation in the legend so that other eagle-eyed readers like our reviewer will no longer have to ponder this omission.   

Reviewer 2. Moreover, why only for cats 6, 11, 15,16, 1821, 22, 24, 25 and 26 are reported the values of Hypergamma globulinemia (before and after treatment)?

Addie et al: Globulin levels were only entered for cats whose globulin levels had not achieved normality (as was explained in the legend) but they have now been put in for all of the recovered cats.

Reviewer 2. Line 174: “This table”: please specify it is table 4.

Addie et al: It was actually the legend for Table 3 so that has been fixed (I hope).

Reviewer 2. Line 174 and table 4. There is incongruity. Table 4 reports data before and after treatment

Addie et al:  I’m sorry, I hadn’t realised that the pdf ran the tables together like that, so line 174 was actually the legend for Table 3, not Table 4. 

Reviewer 2. Table 4: It is not clear the last line “total”

Addie et al: Thank you for pointing this out.  I have changed the name to ‘summary’ and added the following in the legend, hoping to clarify this row:

In the last row a summary of how many cats were anaemic, lymphopenic and hyperglobulinaemic at the first and last samples are shown: FCoV antibody titres were available for 15 of 16 cats and all were high, 13 of 14 last available samples had high antibody titres, one cat’s titre had reduced three-fold from >1280 to 320; 9 of 12 cats were anaemic at the outset, and at the last sample 11 of 13 were anaemic.  Six of 11 cats were lymphopenic rising to 11 of 13; 11 or 13 cats were hyperglobulinaemic at the outset, globulin levels reduced in four cats, probably due to corticosteroids. FIP clinical scores worsened in 3 cats, stayed the same in 5 cats and improved slightly in 5 cats. 

Reviewer 2. Lines 230-232: it is not clear. Please rewrite and clarify the test

Addie et al: I’m sorry that section was unclear and hope that the following sentence makes more sense?

“AGP was not included in the decision to class a cat as recovered or only in remission because the purpose of this study was to assess whether or not AGP reduction was an indicator of clinical recovery.”

Reviewer 2. Figures 1 and 2 are difficult to interpret. There are too many data/lines. They should be redone making them more readable

Addie et al:  Yes, both reviewers found that a problem and it’s something I’ve wrestled with for years.  I thought displaying the graph 3D would help, but excel doesn’t allow the lines to continue in 3D when there are empty cells – it keeps returning to zero . 

I’ve split the recovered cats into 3 graphs (instead of 2 as formerly), separating out those cats treated before and after the introduction of the GS-441524 type analogues, placed the legends onto the chart area to give more space for the actual graphs and reduced the x axis of Figure 2 to 20 months, (from 30): I hope this helps.  I’ve also changed colours of the paler lines to make them more visible. 

The aim of these graphs is to show that the AGP levels of the recovered cats reduced to normal levels, while AGP of cats who are not cured does not: to make this clearer, I’ve put in a dashed red line at the upper normal limit of 500 μg/ml.

Reviewer 2. Lines 319-320: Review the formatting

Addie et al: Thank you, this did appear very oddly in the pdf.  I hope we have fixed the problem now.

Thank you again for your help with our paper.